# Force-Position Hybrid Compensation Control for Path Deviation in Robot-Assisted Bone Drilling

**DOI:** 10.3390/s23167307

**Published:** 2023-08-21

**Authors:** Shibo Li, Xin Zhong, Yuanyuan Yang, Xiaozhi Qi, Ying Hu, Xiaojun Yang

**Affiliations:** 1Guangdong-Hong Kong-Macao Joint Laboratory of Human-Machine Intelligence-Synergy Systems, Shenzhen Institute of Advanced Technology, Chinese Academy of Sciences, Shenzhen 518055, China; sb.li@siat.ac.cn (S.L.); 20s153193@stu.hit.edu.cn (X.Z.); yy.yang1@siat.ac.cn (Y.Y.); xz.qi@siat.ac.cn (X.Q.); ying.hu@siat.ac.cn (Y.H.); 2School of Mechanical Engineering and Automation, Harbin Institute of Technology, Shenzhen 518055, China

**Keywords:** bone drilling, compensation control, stiffness model, robot-assisted surgery, path deflection

## Abstract

Bone drilling is a common procedure in orthopedic surgery and is frequently attempted using robot-assisted techniques. However, drilling on rigid, slippery, and steep cortical surfaces, which are frequently encountered in robot-assisted operations due to limited workspace, can lead to tool path deviation. Path deviation can have significant impacts on positioning accuracy, hole quality, and surgical safety. In this paper, we consider the deformation of the tool and the robot as the main factors contributing to path deviation. To address this issue, we establish a multi-stage mechanistic model of tool–bone interaction and develop a stiffness model of the robot. Additionally, a joint stiffness identification method is proposed. To compensate for path deviation in robot-assisted bone drilling, a force-position hybrid compensation control framework is proposed based on the derived models and a compensation strategy of path prediction. Our experimental results validate the effectiveness of the proposed compensation control method. Specifically, the path deviation is significantly reduced by 56.6%, the force of the tool is reduced by 38.5%, and the hole quality is substantially improved. The proposed compensation control method based on a multi-stage mechanistic model and joint stiffness identification method can significantly improve the accuracy and safety of robot-assisted bone drilling.

## 1. Introduction

With the continuous improvement in orthopedic diagnosis levels and the acceleration of aging processes, the number of orthopedic injuries is increasing and surgery is the main form of treatment. Orthopedic surgical robots have attained increased popularity clinically, with the advantages of high precision, low trauma, less blood loss, and fast recovery [1]. With the continuous development of artificial intelligence, image, sensor, and other technologies, orthopedic surgical robots will have greater autonomous abilities to participate in more and more complex operations; however, their safety is an issue that needs to be progressively focused on [2,3,4].

Bone drilling is an essential and regularly applied surgical procedure in various orthopedic operations, including pedicle screw fixation, lamina decompression, joint replacement, and the internal/external fixation of fractures. It is also one of the most focused operations in robot-assisted orthopedic surgeries for the sake of its robot friendliness [5]. A typical scenario of clinical bone drilling is to develop pilot tunnels, which provide support for the load-bearing screws to transfer loads between the bone tissue and fixator. The primary objective of bone drilling is to meticulously create precise and uncontaminated perforations while safeguarding the integrity of neighboring tissues, thus mitigating the likelihood of adverse outcomes such as nerve impairment, spinal cord trauma, hemorrhage, osseous fracturing, and related complications. However, due to the complex biomechanical characteristics of bone tissue and the undesirable operating conditions of surgical drilling [6,7], accurate bone drilling is still challenging, despite the fact that drilling is an established topic in machining science. The quality of the drilling hole can be assessed based on various indicators, including path deviation, surface finish, accuracy, dimensional consistency, thermal damage, etc. In the context of this study, the path deviation and the tool force are specifically focused on, which indicate the accuracy and reduced tissue disruption during the drilling process. The applications of robots have not been able to alleviate the issue but aggravate it instead.

One of the unsolved issues in bone drilling is drilling path deviation, which may cause poor fixation, unsatisfactory decompression, tool breakage, or even the damage of surrounding tissues, which is extremely dangerous in certain circumstances, as shown in Figure 1. To minimize the risk of drill skidding, surgeons may indent the surface of the cortical bone to enlarge the contact area before drilling inside. In terms of robotic drilling, such indentation action is not easy to realize, while path deviation is mainly contributed by two aspects: end-effector deflection and robot deformation. On one hand, when the drill bit touches the rigid, slippery, and steep cortical surfaces, drill skidding is likely to occur and the highly heterogeneous bone tissue may also cause the drill to deviate from the correct direction [3]. On the other hand, the insufficient stiffness of the robot can significantly reduce the positioning accuracy, which is a well-known issue in the field of robotic machining [8,9]. The two factors combine, causing the deviation of the drilling path, which may result in the damage of the end-effector as well as the surrounding tissue due to the high-speed rotation of the drill.

The analysis of the tool deviation mechanism is of great significance to address the problem. Shu et al. analyzed the mechanism of the bone drill bit slipping, considered tool deflection as result of the unbalanced force of the bit, and improved the deflection problem by designing a new orthopedic bit tool [7]. Yang et al. analyzed the process of tool deflection during machining from the perspective of drilling mechanics and divided the deflection process into multiple stages [10]. Sui et al. divided the force of the drill bit into multiple areas, established a theoretical model of the drill bit in the process of bone drilling, and verified it in a three-stage experimental design [11,12]. Pourkand et al. utilized augmented haptic feedback to simulate the anatomic variability in bone by generating clinically relevant force levels during orthopedic drilling through a predictive model informed by experimental data, achieving accurate force prediction and practical application using a customized Phantom Premium haptic device [13]. Li et al. introduced the gray value of a CT image of bone tissue as a weight into the mechanical model of bone drilling and verified the accuracy of the mechanical model through experiments [14].

In most studies, the low stiffness of the serial robot is considered as the main reason for path deviation [15]. The tool is subject to unbalanced force during the interaction process, which causes the accuracy and stability of the robot to decrease. Bu et al. proposed a Cartesian flexibility model to describe the rigidity of the robot in Cartesian space and defined the quantitative evaluation index of the machining performance [16,17]. Jiao et al. proposed a method for the selection of regular sampling points based on a spatial grid to realize the simultaneous identification of robot kinematics and stiffness parameters [18]. She et al. obtained the maximum working stiffness of the robot drilling system in a machining task by optimizing the installation angle of the electric spindle and the robot end flange [19]. Xu et al. proposed an effective stiffness modeling method for heavy industrial robots considering joint/link flexibility, link mass, and a gravity compensator [20]. Another factor that causes path deviation is the deformation of the robot end-effector. In orthopedic surgery, the length of the drill bit is usually more than 60 mm and the diameter is less than 4.2 mm, which often causes the drill bit to slip along the bone surface [21].

Compensation control methods can be effective in correcting the drilling path. Alici analyzed and described the characteristics of compensation control for robotic drilling, obtained a control strategy according to the characteristics of the task, and completed the drilling operation of a robot manipulator under the control of the contact force [22]. Olsson et al. proposed a method for high-precision drilling using industrial robots with high-bandwidth force feedback, focusing on eliminating the sliding of end-effectors when clamping the workpiece surface [23,24]. Lee et al. determined that the low rigidity of the robot is the main reason for tool tip deviation resulting in machining defects and employed an implicit force-position control approach to correct the deviated path [25]. Yen et al. established a force control approach in a handheld orthopedic robot system to accompany the resolved motion rate controller and overcome tool deflection as a result [26]. Wang et al. proposed real-time compensation methodologies utilizing a single three-degrees-of-freedom laser tracker to enhance the path accuracy and quality in robotic machining operations, leading to significant improvements in hole positioning, path accuracy, and overall machining precision [27]. Zhang et al. introduced innovative methodologies, including stiffness modeling, real-time deformation compensation, and controlled material removal, to enhance the machining performance of flexible industrial robots, achieving improved productivity and surface accuracy, thereby unlocking new practical machining applications [28].

To enhance the safety of robot-assisted bone drilling operations and tackle the issue of path deviation, we present a compensation control method in this paper. Our approach is based on an in-depth mechanical analysis of path deviation, which forms the foundation for the proposed method. In Section 2, the mechanics of the bone drilling process are introduced and several key components are introduced, including the robot stiffness model, the tool deformation model, and the compensation control method. In Section 3, experiments are conducted to validate the proposed mechanical model and compensation method. Section 4 and Section 5 provide a comprehensive discussion of our work and conclusions.

## 2. Materials and Methods

### 2.1. Inclined Drilling Mechanics

To analyze the general mechanical characteristics of tool cutting in the bone drilling process, the classical twist drill model is applied in our work and the mechanistic modeling approach is implemented. To set up a clear reference frame for our analysis, both a global coordinate system XYZ and a local coordinate system xyz for the end-effector are established, as depicted in Figure 2a,b. The local coordinate system xyz is fixed to the drill bit and rotates along with it, while the global coordinate system XYZ is stationary relative to the robot flange. When the drill bit first makes contact with the bone surface, the two coordinate systems are aligned with each other.

The idea of discrete summation is adopted for the drilling cutting force modeling [29,30]. By discretizing the cutting process into small elements, the cutting force of the tool can be accurately modeled during the drilling operation. According to Figure 2c, the tool element force can be formulated as
(1)$$\left[\begin{array}{c} dF_{tc}\\ dF_{fc}\\ dF_{rc} \end{array}\right]=db\cdot h\left[\begin{array}{c} K_{tc}\\ K_{fc}\\ K_{rc} \end{array}\right]$$
where $dF_{tc}$, $dF_{fc}$, and $dF_{rc}$ represent the tangential, axial, and normal force elements, respectively. $K_{tc}$, $K_{rc}$, and $K_{fc}$ are the tangential, axial, and normal force cutting coefficients, respectively. *h* represents the cutting thickness and db represents the length of the discrete element along the cutting edge.

The element forces in the coordinate system of the cutting edge need to be converted into xyz coordinates. The conversion can be achieved through the transformation matrix T as follows:(2)$$\left[\begin{array}{c} dF_{x}\left(z\right)\\ dF_{y}\left(z\right)\\ dF_{z}\left(z\right) \end{array}\right]=T\left[\begin{array}{c} dF_{fc}\\ dF_{tc}\\ dF_{rc} \end{array}\right]$$

In orthopedic surgery, the feed direction of the drill bit cannot be exactly perpendicular to the bone surface in most cases due to the limitations of the working space. This results in an inclined interaction between the drill bit and the bone surface, with the angle between the tool and the bone tissue denoted by β, as shown in Figure 3. This inclined interaction causes the force on a single cutting edge of the tool to change in a quasi-periodic manner throughout the drilling process. The reason for this phenomenon is that the actual cutting length between the cutting edge and the bone tissue (the length of the line intersecting the cone with the plane) changes in a quasi-periodic manner. The interaction angle β leads to unequal force areas on the two cutting edges of the tool, resulting in unbalanced forces in the *X* direction in the global coordinate.

The whole process of inclined drilling is divided into four stages, as shown in Figure 3: (1) the left cutting edge is locally loaded; (2) the left and right cutting edges are locally loaded; (3) the left cutting edge is fully loaded and the right cutting edge is locally loaded; (4) the left and right cutting edges are fully loaded. Due to the different sizes of the force area, the force is generally unbalanced in Stages 1, 2, and 3, making it highly likely for the tool path to deviate with insufficient surface friction. However, in Stage 4, the whole drill bit has entered the bone tissue, resulting in equilibrium. According to the drill geometry, the time spent in each active stage can be calculated as
(3)$$\begin{array}{cc} t_{1} & =\frac{2x_{0}\tan\beta }{c} \end{array}$$
(4)$$\begin{array}{cc} t_{2} & =\frac{z_{1}\sin\frac{\pi }{2}-k_{t}-\beta }{c\cos\beta \cos k_{t}} \end{array}$$
(5)$$\begin{array}{cc} t_{3} & =\frac{\left(x_{1}-x_{0}\right)\tan\beta +z_{1}}{c} \end{array}$$
in which $x_{0}$, $x_{1}$, and $z_{1}$ are the geometric parameters of the drill bit, as shown in Figure 2a. $k_{t}$ is the angle related to the geometry of the drill bit. *c* is the tool feeding rate.

Integration is performed on the transformed element forces, and the unbalanced deflection force in the X direction in the global coordinate system can be written as
(6)$$F_{X}=\left\{\begin{array}{cc} \int_{0}^{ct}dF_{x} & \;\mathrm{if}\;0\leq t\leq t_{1}\\ \int_{0}^{ct}dF_{x}-\int_{0}^{c(t-t_{1})}dF_{x} & \;\mathrm{if}\;t_{1}<t\leq t_{2}\\ \int_{0}^{ct_{2}}dF_{x}-\int_{0}^{c(t-t_{1})}dF_{x} & \;\mathrm{if}\;t_{2}<t\leq t_{3} \end{array}\right.$$

### 2.2. Stiffness Modeling

In addition to the unbalanced force on the tool, the insufficient stiffness of the robot can also cause positioning errors, which can lead to path deviations in bone drilling, as illustrated in Figure 4. Therefore, it is important to comprehend the stiffness properties of orthopedic surgical robots and cutting tools. This section presents a description of the theoretical derivation and experimental identification principles for the robot stiffness model, followed by an analysis of the mechanical deformation of the surgical tool.

#### 2.2.1. Stiffness Model of Robot

The stiffness of a robot mainly is mainly contributed by the links and the joints. Due to the fact that the deformation at joints is much larger than the deformation at links, it is assumed that the links are rigid and only the joints are deformed, as shown in [14]. Therefore, the robot stiffness can be evaluated from the joint stiffness as follows.
(7)$$K=C^{-1}=J^{-⊤}K_{\theta }J^{-1}$$
where K is the robot stiffness, C is the robot compliance, J is the robot Jacobian matrix, and $K_{\theta }$ is the robot joint stiffness. From this, the mapping relationship between the stiffness of the robot joints and the stiffness of the robot end can be obtained. Experimental identification of the joint stiffness is performed to evaluate the total stiffness according to Equation (Equation 7).

Within the bone drilling process, the torsion components on the end-effector are negligible, while the translational displacement can be obtained through optical measurements. According to Equation (Equation 7), the displacement and the joint compliance are related through
(8)U=BD
where U is the displacement vector, D is the rearranged joint stiffness. U, and D and B are defined as
(9)$$\begin{array}{cc} U & ={\left[\begin{array}{ccc} U_{1} & U_{2} & U_{3} \end{array}\right]}^{⊤} \end{array}$$
(10)$$\begin{array}{cc} D & ={\left[\begin{array}{cccccc} K_{\theta 1}^{-1} & K_{\theta 2}^{-1} & K_{\theta 3}^{-1} & K_{\theta 4}^{-1} & K_{\theta 5}^{-1} & K_{\theta 6}^{-1} \end{array}\right]}^{⊤} \end{array}$$
(11)$$\begin{array}{cc} B & =\left[\begin{array}{ccc} J_{11}\sum_{i=1}^{6}J_{i1}F_{i} & \cdots & J_{16}\sum_{i=1}^{6}J_{i6}F_{i}\\ \vdots & \ddots & \vdots \\ J_{61}\sum_{i=1}^{6}J_{i1}F_{i} & \cdots & J_{66}\sum_{i=1}^{6}J_{i6}F_{i} \end{array}\right] \end{array}$$

By adjusting the pose of the robot and the force at the robot flange, multiple pairs of X and B can be obtained. Using Equation (Equation 8), the compliance C can be evaluated numerically.

#### 2.2.2. Stiffness Model of Tool

When the drill bit encounters a non-vertical cone surface, it tends to deviate from the desired drilling path. This can be attributed to the improper control of applied force and contact force situations. When drilling through hard and slippery bone surfaces, excessive force may cause the drill bit to slip. Assuming that the cutting tool is rigid, the expected path of the surgical robot may deviate significantly from the actual path. This can negatively affect the compensation control of the robot path as it does not consider the deformation of the tool. To address this issue, it is necessary to develop a stiffness model for the tool.

The tool can be deflected when forces and torques are applied in bone drilling operations. The cantilever beam model is used to model the deformation of deflection. According to the coordinate system shown in Figure 5, the stiffness of the tool can be written as
(12)$$k=\left[\begin{array}{ccccccc} \frac{3EI_{x}}{l^{3}} & \frac{3EI_{x}}{l^{3}} & \frac{3EI_{y}}{l^{3}} & \frac{A_{z}E}{l} & \frac{2EI_{x}}{l^{2}} & \frac{2EI_{y}}{l^{2}} & \frac{GI}{l} \end{array}\right]$$
where *l* is the length of the tool, *E* is the Young’s modulus of the tool, *G* is the shear modulus of the tool, $I_{x}$ and $I_{y}$ are its moments of inertia in the *x* and *y* directions, and $A_{z}$ is the cross-section area of the tool.

### 2.3. Compensation Control

In robot-assisted bone drilling, the end of the tool is subjected to large forces. The insufficient combined stiffness of the serial robot and the stiffness of the surgical tool will result in the deviation of the tool path. In this section, the force-position hybrid compensation control of the deviation during robot-assisted surgery will be carried out in combination with the stiffness model established earlier. A control framework of force-position hybrid control is proposed, as shown in Figure 6. As noted in Figure 3, the *X* direction in the global coordinate system is the main deviation direction, and compensation control is performed in this direction.

#### 2.3.1. Force-Position Hybrid Control Method

In the bone drilling process, the torsional deviation of the tool path can be ignored as the torque exerted on the tool by bone tissue is very small. The main factor affecting the quality of bone drilling is the positional deviation of the tool path. Therefore, in order to simplify the problem, this paper mainly focuses on the relationship between the positional deviation of the tool and the force exerted on it. Position deviation is mainly composed of robot positioning error and tool deformation.

In the interaction between the tool and the bone tissue slope, the tool is prone to producing large deformation errors. In this compensation method, the trajectory deviation caused by this error is first corrected offline, and the displacement of the tool is obtained by the force model of the drill and the stiffness model of the tool: (13)$$\Delta P_{x}=k_{D}^{-1}F$$
where $k_{D}$ is the first three columns in k, which represent the deflection factors, and $F={\left[\begin{array}{ccc} F_{x} & 0 & 0 \end{array}\right]}^{⊤}$.

If the trajectory is simply modified offline, the effect of compensation is not satisfactory. Moreover, the positioning error of the robot in the surgical process has not been compensated for. On this basis, online compensation control is carried out. The robot positioning error is expressed as
(14)$$\Delta x_{1}=C_{D}^{-1}T_{1}F_{f}$$
in which $C_{D}$ is the simplified compliance matrix, which includes the values of the first three rows and the first three columns in the original compliance matrix C. $T_{1}$ is the transformation matrix between the robot base coordinate system and the sensor coordinate system. $F_{f}={\left[\begin{array}{ccc} F_{xf} & 0 & 0 \end{array}\right]}^{⊤}$ and $F_{xf}$ is the filtered value of the sensor.

The deformation of the surgical tool can be determined as
(15)$$\Delta x_{2}=T_{2}K_{FD}^{-1}F_{f}$$
where $T_{2}$ is the transformation matrix between the local coordinate system of the tool and the robot base coordinate system. Therefore, considering these two contributions, the deviation of the tool path to be compensated is
(16)$$\Delta x=\Delta x_{1}+\Delta x_{2}$$

In this method, the unbalanced force is measured by a force sensor. Then, according to the stiffness of the robot and the stiffness of the tool, the error is converted into a position signal, which compensates for the error by adding it to the motion controller of the robot. This method does not need to switch between force and position control, so it is more stable than force-based impedance control.

#### 2.3.2. Compensation Strategy

Since the robot compensation takes place after the deformation of the end-effector, the online compensation approach is expected to have hysteresis behavior, which mainly arises from two aspects: the time required for data acquisition and transmission, and the time required for the robot to execute error compensation commands. According to the work by Shi et al., the time required for data acquisition and transmission in an industrial robot is approximately 2 ms, and the time for robot compensation error is approximately 12 ms [31]. Therefore, the overall lag time is on the scale of 14 ms. In addition, since the data acquisition speed is faster than the robot compensation speed, the accumulation of compensation commands may result in oscillation. To overcome the above issues, a PID controller is adopted to reduce the oscillation, while, at the same time, the feedback and feedforward control methods are also applied in the proposed system, which can effectively reduce the influence of the delay.

The difference between the deviations of the two adjacent points, *i* and *j*, on the trajectory, is given by
(17)$$\Delta d_{i}=\Delta x_{i}-\Delta x_{j}$$
in which $\Delta x_{i}$ represents the deviation of the tool at point *i* on the trajectory. When the difference value is obtained, the compensation of the robot tool path will be calculated according to PID control: (18)$$u_{i}=K_{P}\Delta d_{i}+K_{I}\sum \Delta d_{i}+K_{D}\left(\Delta d_{i}-\Delta d_{i-1}\right)$$
where $K_{P}$, $K_{I}$, $K_{D}$ are proportional, integral, and differential coefficients, respectively.

To achieve the ability to modify its path in real time, a path prediction control strategy is proposed, as depicted in Figure 7. At point *i* of the motion trajectory, the robot is executing the motion command for point $P_{i+1}$. To send the compensation value to the correction generator, $u_{i}$, which was calculated at point *i*, can be applied to the correction generator at point $P_{i+2}$. As the tool moves to the next point, $P_{i+2}$, a new target point, $P_{i+2}^{\prime }$, is determined by adding the calculated compensation value, $u_{i}$, to the original target point position information, $P_{i+2}$. The next new target point, $P_{i+3}^{\prime }$, is obtained in a similar manner. During each movement, the target point information is updated with the new target point information obtained by adding the compensation value to the original target point information. The compensation value is calculated by using the collected current force information and the robot posture information, and it can only be added to the target point information at alternate points. Meanwhile, the master computer constantly collects real-time information and calculates the compensation value, enabling the robot to update its compensation value and correct its path.

In order to realize the dynamic adjustment of its trajectory in real time, an innovative path prediction control strategy is introduced, as elucidated in Figure 7. This strategic framework enables the robot’s motion to adapt to unforeseen circumstances, enhancing the precision and flexibility throughout its operation.

At a given juncture *i* along the trajectory, the robot is in the process of executing a motion command targeting point $P_{i+1}$. To facilitate the application of corrective measures, the compensation value $u_{i}$, previously computed at point *i*, is forwarded to the correction generator, subsequently impacting the motion at point $P_{i+2}$. As the tool progresses to the subsequent point, $P_{i+2}$, a new target point, $P_{i+2}^{\prime }$, is determined by incorporating the calculated compensation value, $u_{i}$, with the initial position data of $P_{i+2}$. This iterative process continues, defining the new target point, $P_{i+3}^{\prime }$, in a similar fashion. Through each motion segment, the target point information is continually refreshed using the recalculated target point augmented with the compensation value. Calculating this compensation value involves an amalgamation of real-time force data and the robot’s posture information, contributing to the system’s ability to adjust its path. It is important to note that the application of the compensation value exclusively takes place at alternate points, minimizing potential disturbances to the robot’s motion.

Simultaneously, a master computer consistently gathers and processes real-time information, facilitating the continual calculation of the compensation value. This iterative feedback loop empowers the robot to iteratively update its compensation value and fine-tune its trajectory to ensure accurate and adaptive motion.

In summary, the path prediction control strategy embodies a sophisticated mechanism by which the robot can dynamically correct its trajectory by systematically integrating real-time force feedback and posture information. This intricately orchestrated process underscores the adaptability and precision of the robot’s motion, culminating in enhanced performance and trajectory modification capabilities.

## 3. Results

### 3.1. Simulation Results

In this study, a specific type of drill bit was chosen and its geometric parameters, interaction angle, and feed rate were determined. A semi-static finite element analysis (FEA) was employed to simulate the mechanical condition of the tool at the initial contact, and the results are illustrated in Figure 8a. The drilling tool is modeled as an elastic model with a Young’s modulus of 210 GPa and Poisson’s ratio of 0.3, while the bone is modeled as a rigid surface with a friction coefficient. According to the simulation, the drilling tool will deflect when in contact with the bone surface, resulting in reaction forces in the horizontal direction. Different friction coefficients may result in different horizontal reaction forces, which can characterize the deflection behavior, as shown in Figure 8b. From Equation (Equation 6), the unbalanced reaction force can also be evaluated for the entire drilling process, and the results are illustrated in Figure 9. The curve shows that the unbalanced force increases initially with the feed of the drill bit and then decreases, with the maximum unbalanced force occurring during Stage 2. In Figure 9b, the comparison experiment curve shows that the force values during Stages 1 and 2 are consistent with the actual situation, with both reaching similar maximum force values. Furthermore, the force value remains constant at the maximum level during Stage 3, indicating that the tool path has deviated, resulting in continuous loading from the bone tissue.

From Equation (Equation 6), it is evident that the unbalanced deflection force is dependent on the interaction angle and the robot feed rate during the first three drilling stages. Our primary focus is on determining the maximum force that causes deviation. By analyzing the unbalanced force for each drilling scenario, the force causing deviation can be evaluated. The maximum force in each simulation helps to establish a relationship between the unbalanced force, the interaction angle, and the robot feed rate. As shown in Figure 10a, the maximum unbalanced force is roughly linearly related to the interaction angle when the angle is less than 20°. For angles greater than 20°, the maximum unbalanced force remains almost constant. The 20° angle is primarily related to the geometric model of the drill bit and is typical for most twist drills. Figure 10b illustrates that the maximum unbalanced force is approximately linearly proportional to the robot feed rate when the interaction angle is fixed.

During orthopedic surgery, on the one hand, the interaction between the drill and bone tissue is usually inclined due to the relatively small operating space. On the other hand, in order to control the operation time, the feed rate cannot be too small. Therefore, in the process of robot-assisted bone drilling, the drill bit is affected by the unbalanced force, resulting in the deviation of the tool path. This deviation will further reduce the quality of the hole in the surgery, leading to asymmetry or roughness.

### 3.2. Experimental System

The experimental setup for the study is depicted in Figure 11. In the experiment, the UR5 (CB3) universal robot was used as the experimental carrier, with a maximum load of 5 kg. The UR robot’s versatility, user-friendly programming, precision, and cost-effectiveness make it well-suited for medical research experimentation, enabling the rapid prototyping of concepts and interdisciplinary collaboration at a lower cost compared to specialized medical robots. The force needed for bone drilling is typically less than 10 newtons, which is far less than the maximum load of the UR robot. A straight surgical drilling tool was attached to the end-effector of the UR5. The force/torque sensor was calibrated and fixed between the robot and the drilling tool. Neatly cut buffalo bone specimens were employed as our experimental subjects, on which drilling procedures were conducted to simulate surgical drilling scenarios for human bone. Another reason for utilizing this experimental material was its standardized cutting shape and dimensions, ensuring excellent experimental reproducibility. The specimens were securely fixed to the experimental platform. The NDI optical tracker was utilized for the deviation measurement.

The control box of the UR5 robot system was connected to the upper computer via Ethernet. An ATI six-axis sensor (ATI-Mini40-SI-40-2) with a resolution of 1/100 N in Fx/Fy and 1/50 N in Fz was installed on the flange of the UR5 universal robot to detect the real-time force during orthopedic surgery, which was then transmitted back to the upper computer through a data acquisition box. Additionally, the NDI optical tracking system was connected to the upper computer through a USB. The experiment aimed to verify the effectiveness of the control method by detecting the deviation of the tool path during orthopedic surgery.

### 3.3. Robot Joint Stiffness Identification

Following the robot joint stiffness identification approach, several groups of experiments were conducted. The robot joint movement, the load, and the tracking data of the NDI optical tracker were recorded, respectively. The force, expressed under the base frame, was consistent with theoretical derivation. The conversion between the NDI frame and the robot coordinate system was calibrated and carried out. The deviation in the experimental data was transformed into the base frame and then calculated using the proposed joint stiffness identification strategy. As a result, the joint stiffness values were obtained as $diag[2.53\times 10^{4},2.18\times 10^{4},1.62\times 10^{4},7.23\times 10^{4},6.62\times 10^{4},4.43\times 10^{4}]$ for each value with the unit of N·m/rad.

The NDI optical tracker was used to measure the actual deviation, and the theoretical deviation caused by robot stiffness was calculated using the robot joint stiffness values. Three different postures were applied in the experiment, and the results are presented in Table 1. The primary deviation direction was the *Z*-axis, and the results indicated an error within 0.15 mm, demonstrating the accuracy of the robot joint stiffness.

### 3.4. Compensation Results

According to the compensation control method proposed above, the verification experiment was carried out. According to the analysis in Section 2, the interaction angle was set to 30°, and the feed rate was 0.5 mm/s. Drilling with and without deviation compensation control were carried out for comparison. The main parameters applied in the experiment are given in Table 2.

In order to achieve online compensation for robot-assisted bone drilling operations, it is essential that the time required for each function in the upper computer software meets real-time requirements. Specifically, the time required to obtain the robot’s position and pose, calculate the compensation amount, and transmit the compensation amount to the internal variables of the robot controller in the upper computer must be within 50 ms. The sampling frequency of the force sensor was 100 Hz, and 20 points were selected for each force processing for filtering, resulting in a sampling time of 200 ms for the milling force in the upper computer. Therefore, the time for the system to complete the compensation once was less than 250 ms. The time for each trajectory point of the robot was 600 ms, which further confirmed that the compensation scheme satisfied the real-time requirements.

The results of the compensation control method are presented through three data aspects: tool force, tool path deviation, and hole quality. The first aspect, tool path deviation, is illustrated in Figure 12, which displays the deviation value of the NDI target point in the X direction of the inertial coordinate system during the tool feeding process. To highlight the intricate relationship between the KP value and the mechanical properties of the tool and robot joint, the tool deviation states are plotted against the proportional compensation parameters. As depicted, the scale coefficient plays a crucial role in reducing the deviation amount. The force of the tool was monitored using the ATI six-axis force sensor during the robot bone drilling process, shown in Figure 13. The curve in Stage b reflects the unbalanced force that causes the tool path deviation, while Stage c indicates stable drilling. The results show that a lower load can be achieved in the process of bone drilling using the proposed compensation control scheme, which may benefit the drilling operation in terms of safety, accuracy, difficulty level, and also the endurance of the end-effector.

Finally, the qualities of the holes from the two groups of experiments are shown in Figure 14. From Figure 14a, it can be seen that without effective compensation control, the center of the hole is significantly deviated from the expected position and the quality of the hole is poor. In contrast, the quality of the hole is substantially improved by drilling with the proposed compensation control strategies, as shown in Figure 14b.

## 4. Discussion

In this study, we investigated the force model of a tool at the end of a surgical robot and its influencing factors during the process of bone drilling. A compensation control method was proposed based on the robot stiffness model, tool stiffness model, and tool force to correct tool path deviation and improve the path accuracy in bone drilling.

The most direct factors causing path deviation during bone drilling are force and stiffness, and our method directly addresses these factors. The interaction force during bone drilling is an important factor causing path deviation, which is closely related to the drill parameters, robot parameters, and interaction state. In our experiments, the above parameters were kept consistent to ensure the accuracy of the mechanical model. However, the bone drill parameters can be further studied as a separate component in future research.

The compensation method comprises two parts: the tool force model and the tool stiffness model. Offline correction of the operation path is performed by combining the two models and then online path compensation is performed during surgery by combining the implementation force values collected by sensors and the robot stiffness model. The combination of offline and online compensation effectively improves the accuracy. Our compensation control method offers an accessible and reliable scheme to implement in the actual surgical process.

The experimental results indicate that our compensation method is effective in correcting tool path deviation. The effect of the compensation control method on tool path deviation was analyzed using the NDI tracking device to indicate the deviation state of the tool path. The average deviation value was 0.703 mm without compensation control, 0.587 mm with compensation control, and 0.305 mm with an appropriately increased scale coefficient ($K_{P}=4.0$). The proportional control parameter can speed up the system response and improve the accuracy of path compensation control. However, to ensure system stability, the proportional system cannot be infinitely increased.

The proposed compensation control method significantly reduces the offset force in the interaction between the tool and bone tissue. During drilling, sudden contact between the tool and bone tissue is the greatest cause of this phenomenon. The maximum value of the force was 1.3 N without compensation control, and it was reduced to 0.7 N with compensation control, a reduction of 38.5%. The pilot hole can play a role in reducing the slipping of the drill bit and lessening the deviation problem. The stage of stable drilling had an average force of 1.2 N without compensation control and was 0.3 N with compensation control, a decrease of 75%. The reduction in force ensures the stability of the interaction and reflects the reduction in the deviation of the tool path.

Our compensation control method also greatly improves the hole quality. Without compensation control, the hole quality was rough, with significant defects, and the hole center position deviated from the expected position by 0.8 mm. After using force and position compensation control, the quality of the hole wall was significantly improved, with no defect at the entrance of the hole, and the deviation of the hole center was only 0.2 mm. Therefore, our proposed force and position compensation control can improve the deviation of the tool path.

Despite the promising results in improving the accuracy of bone drilling, there are certain limitations to consider in this research. The compensation method only addresses the factors of force and stiffness in causing path deviation, while other factors, such as the texture of the surface, vibration, and temperature, may also play a role in affecting the accuracy of the drilling process. We acknowledge the assumption of consistent interaction forces during drilling, potentially disregarding variations in bone tissue density and hardness across regions. To address this, our future research will comprehensively explore the impact of these intricate bone properties and biomechanical models. Our forthcoming investigations will examine how diverse bone properties influence the drilling process, shedding light on how fluctuations in density and hardness within bone tissue may affect the interaction forces. By delving into these factors, we aim to enhance the compensation methods, thereby improving the accuracy in bone drilling for surgical robot applications. Additionally, we are intrigued by innovative techniques such as neural network algorithms in refining robot compensation control. In summary, our ongoing efforts will focus on studying bone properties and biomechanical models to optimize the compensation methods and advance the accuracy of bone drilling processes in surgical robot applications.

## 5. Conclusions

This study addresses bone drilling path deviation by identifying key contributors such as tool deformation and robot dynamics. We introduce a mechanical model for inclined drilling force calculation, a unique stiffness identification method for robot joints, and an integrated evaluation of tool and robot deformations. Our proposed force-position hybrid control strategy, based on predictive path modeling, effectively compensates for real-time path deviation, enhancing the drilling accuracy by mitigating tool deflection and robot stiffness limitations. This research contributes refined mechanical insights and a sophisticated control approach, promising improved precision in robot-assisted orthopedic surgery for enhanced surgical outcomes.

## Figures and Tables

**Figure 1 sensors-23-07307-f001:**
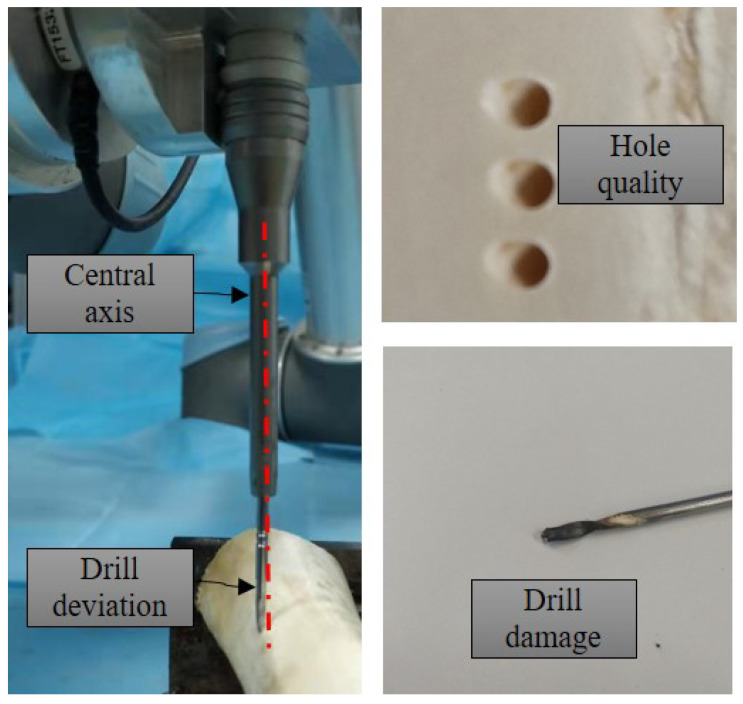
Example of drill deflection resulting in drilling path deviation, poor quality of hole, and drill bit damage.

**Figure 2 sensors-23-07307-f002:**
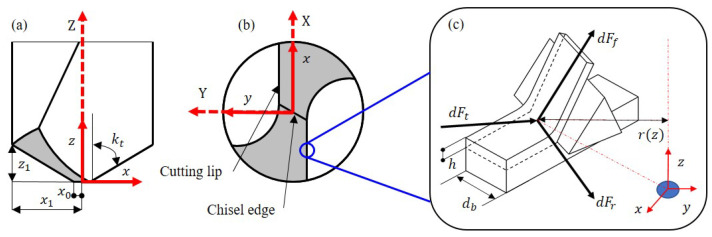
Geometric structure and unit force of fried dough twist drill. (**a**) Side view and coordinates. (**b**) Bottom view and coordinates. (**c**) Force analysis of micro-element on cutting edge. The tangential, radial, and axial shear forces are denoted by $dF_{t}$, $dF_{r}$, and $dF_{f}$, respectively.

**Figure 3 sensors-23-07307-f003:**
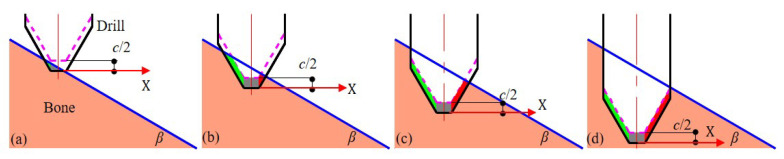
Four stages of inclined drilling. (**a**) represents the local loading of the left cutting edge, (**b**) represents the local loading of both the left and right cutting edges, (**c**) represents the full loading of the left cutting edge and local loading of the right cutting edge, and (**d**) represents the full loading of both the left and right cutting edges. During one cycle, the green area denotes the left force area and the red area denotes the right force area. The feed rate is denoted by *c*, and β represents the interaction angle.

**Figure 4 sensors-23-07307-f004:**
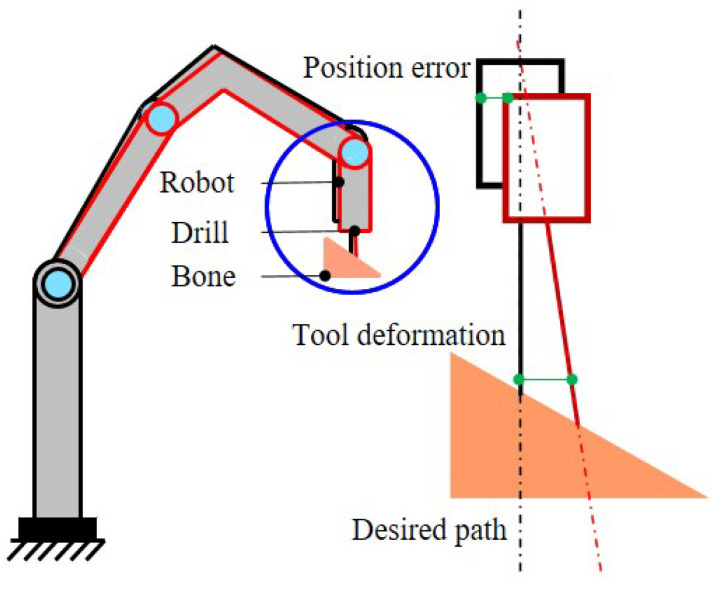
Deviation in robot-assisted bone drilling operation: robot positioning error and tool deformation.

**Figure 5 sensors-23-07307-f005:**
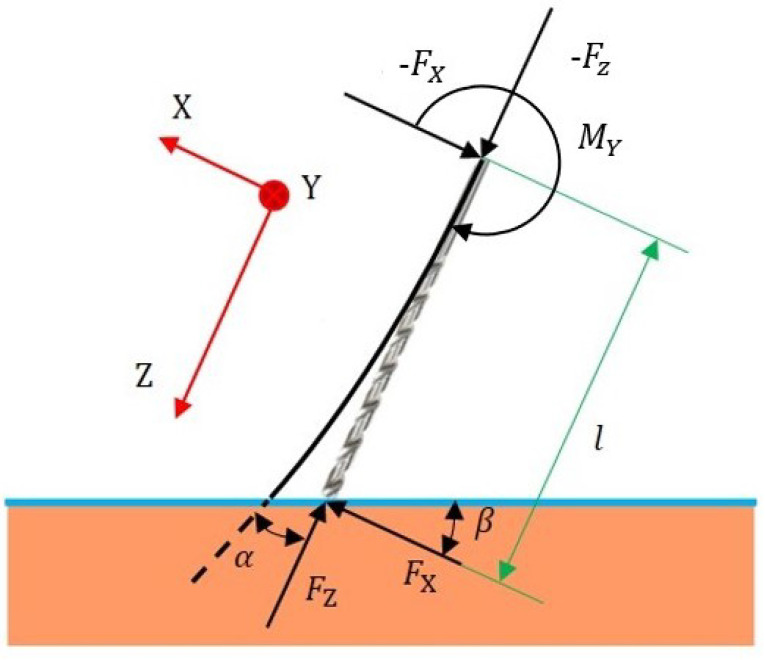
The cantilever beam model of the drilling tool.

**Figure 6 sensors-23-07307-f006:**
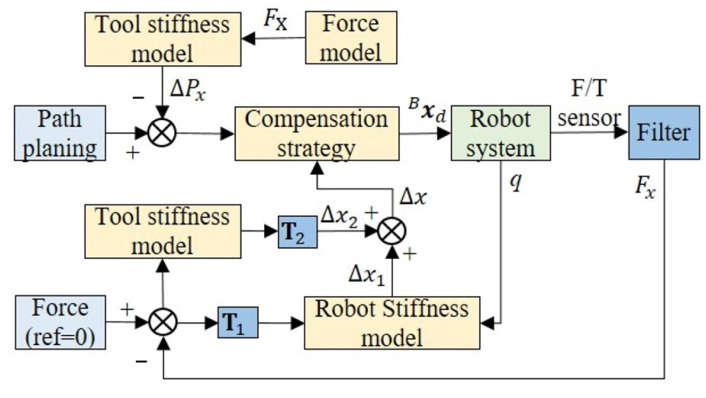
A block diagram of the robot force and position control system, including offline control based on the force model and tool stiffness, and online control based on the robot stiffness and tool stiffness. ${}^{B}x_{d}$ is the pose under the base coordinate and *q* denotes the joint coordinates of the robot arm.

**Figure 7 sensors-23-07307-f007:**
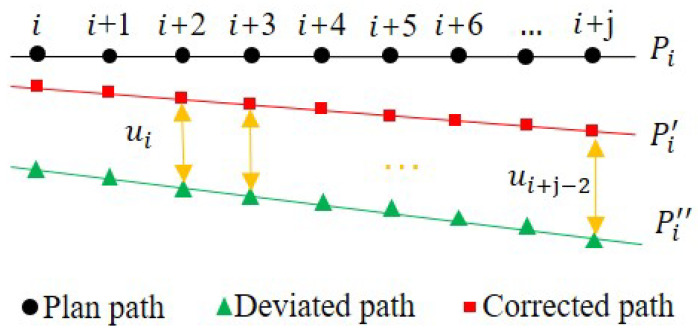
Schematic diagram of path prediction strategy.

**Figure 8 sensors-23-07307-f008:**
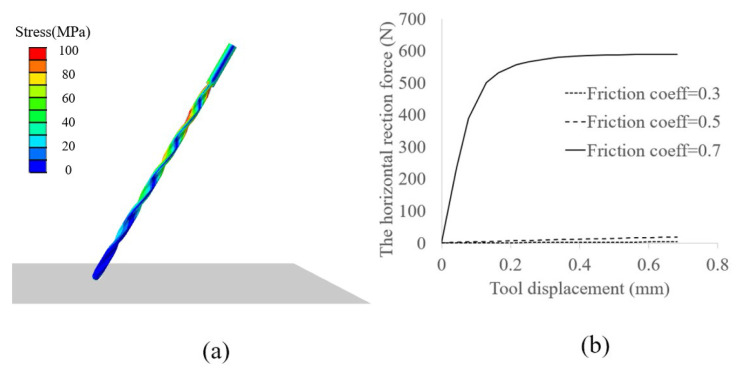
Finite element simulation of the initial deflection process. (**a**) FEA model of the tool moving towards the feeding direction, (**b**) horizontal reaction force changes with tool displacement.

**Figure 9 sensors-23-07307-f009:**
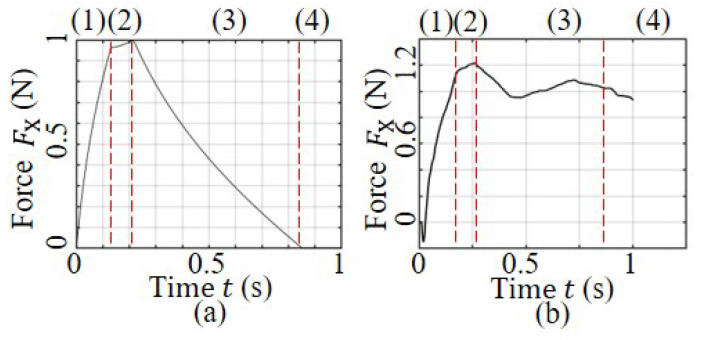
Force–time curve under different stages: (**a**) predicted by model; (**b**) obtained by experiment.

**Figure 10 sensors-23-07307-f010:**
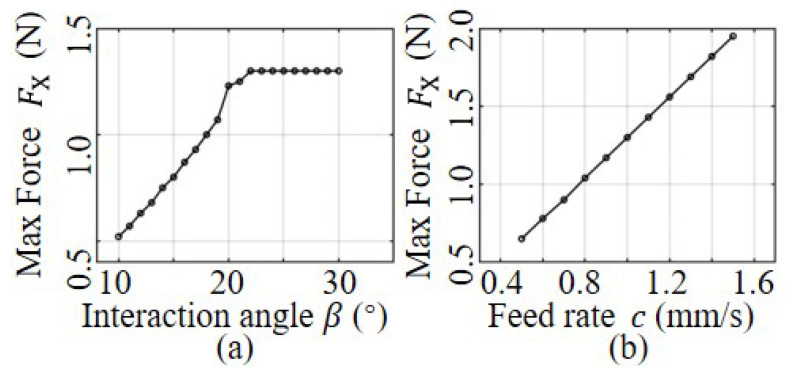
Analysis of mechanical model parameters: (**a**) offset force vs. interaction angle; (**b**) offset force vs. feed rate.

**Figure 11 sensors-23-07307-f011:**
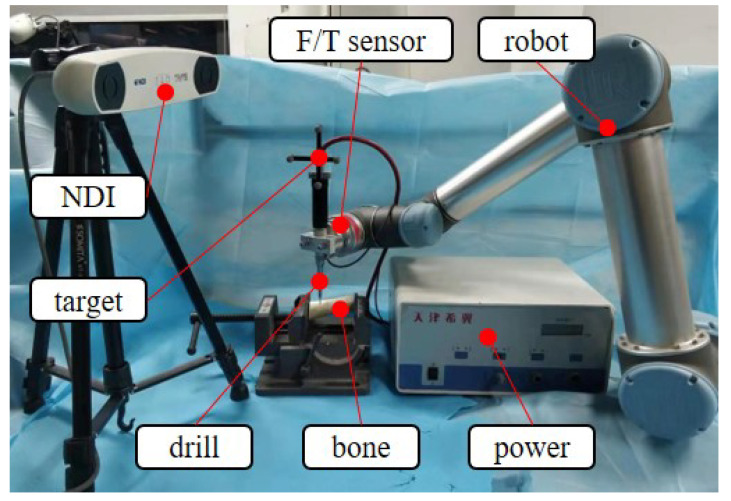
Experimental setup.

**Figure 12 sensors-23-07307-f012:**
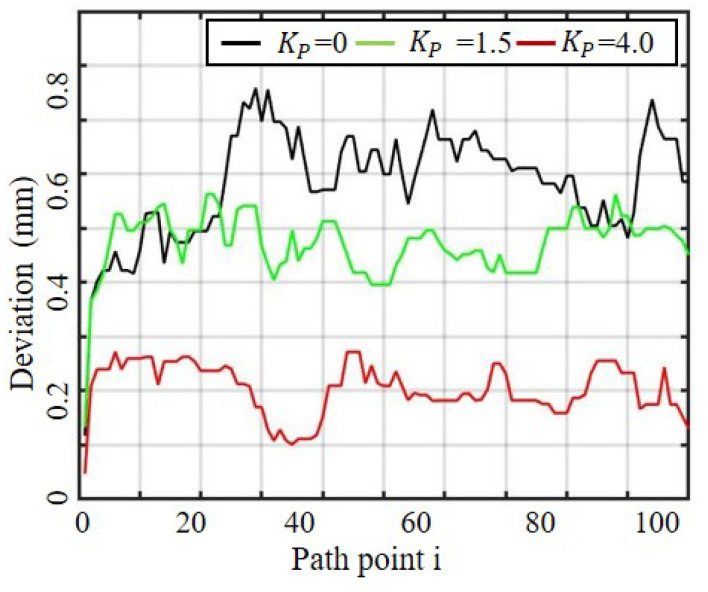
The measured deviation under different proportional compensation schemes.

**Figure 13 sensors-23-07307-f013:**
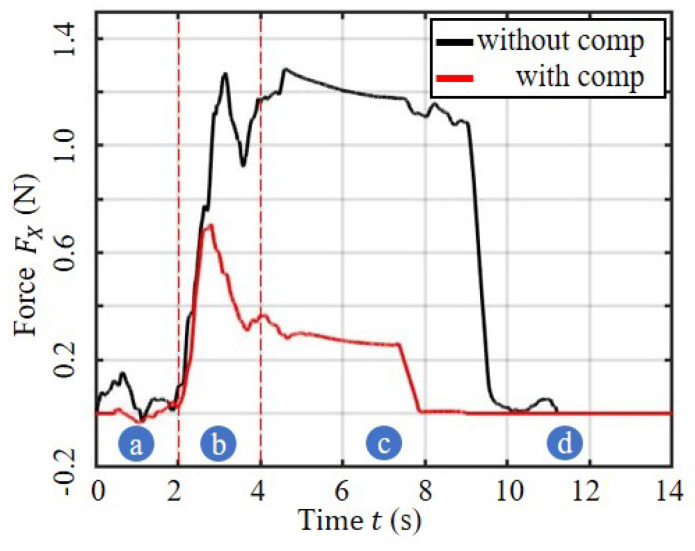
The deflection force when drilling with and without compensation. Four stages are shown: (**a**) the preparation stage, (**b**) the interactive stage, (**c**) the stable drilling stage, and (**d**) the end stage.

**Figure 14 sensors-23-07307-f014:**
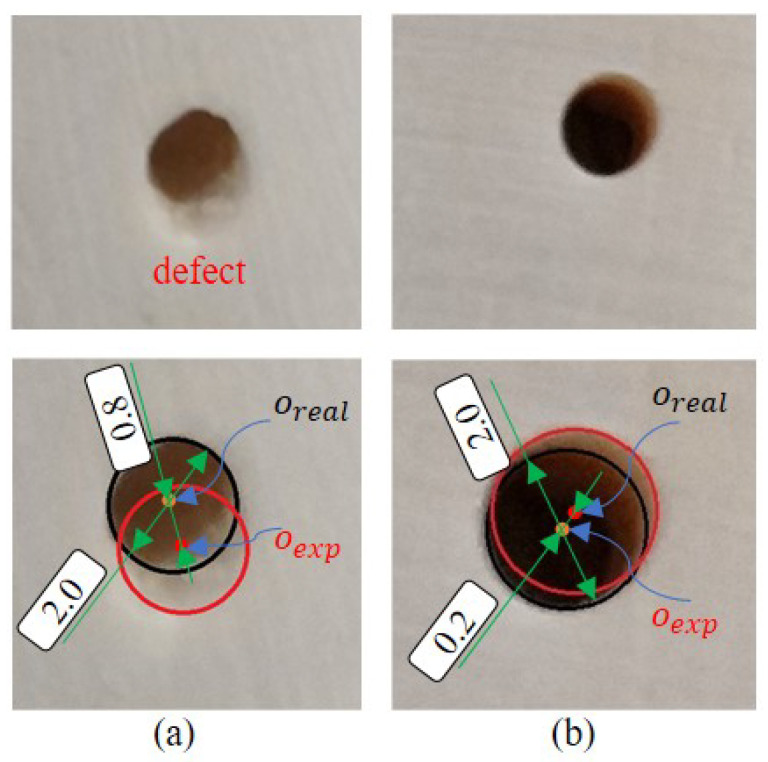
Comparison of hole quality without compensation (**a**) and with compensation (**b**). $o_{exp}$ is the expected position center of the hole; $o_{real}$ is the actual position center of the hole. Unit: mm.

**Table 1 sensors-23-07307-t001:** Comparison of measured and calculated deviation due to robot stiffness.

	No.	Experimental Value (mm)	Theoretical Value (mm)	Error (mm)	Mean Error (mm)
	1	0.04	0.03	0.01	
x	2	−0.15	−0.02	0.13	0.06
	3	0.36	0.31	0.05	
	1	0.08	−0.010	0.18	
y	2	−0.10	−0.00	0.10	0.10
	3	0.12	0.08	0.04	
	1	−0.62	−0.54	0.08	
z	2	−0.81	−0.56	0.25	0.15
	3	−1.11	−0.99	0.12	

**Table 2 sensors-23-07307-t002:** Experiment parameters.

Parameter	Value
Tool type	Twisted drill
Diameter	1.5 mm
Lip number	2
Helix angle	28°
Rotation	20,000 rpm
Feed rate	1 mm/S
Sampling frequency	100 Hz

## Data Availability

Not applicable.

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
