# Peer review of "Force-Position Hybrid Compensation Control for Path Deviation in Robot-Assisted Bone Drilling"

_sensors, 2023, doi:10.3390/s23167307_

Round 1

Reviewer 1 Report

The article addresses the problem of tool path deviation due to the deformation of the tool and the robot in bone drilling surgery, establishes a stiffness model of the robot and the drilling tool, and based on the derived model and the compensation strategy of path prediction, proposes a force-position hybrid compensation control framework to compensate for the path deviation in robot-assisted bone drilling. And the effectiveness of the method is verified by experiments, and the research in the article can improve the accuracy and safety in robot-assisted bone drilling surgery. The research content has certain practical value, but some details of the paper still have to be pushed and improved. The details are as follows:

1.     What do the parameters BXd and q appearing in Figure.8 stand for, and secondly, how the mathematical modeling of the robotic system is done, needs to be explained.

2.     Regarding the force-position hybrid compensation control, many scholars have already carried out research, and the authors are advised to read more about the existing results. The authors are advised to explain how the force-position hybrid compensation control proposed in the paper is innovative compared with the previous research results.

3.     What is the experimental object used by the authors in the experiment, a real skeleton? If it is a real skeleton, the maximum load of 5kg for the UR5 robot, is it able to fulfill the requirement, and it is suggested to state it.

4.     The selection of the KP value will affect the stability of the system, Figure.11 in the three KP value of the selection of the basis, PID control in the KI, KD on the experimental results of the impact of how, it is recommended that give an explanation of the description.

Reviewer 2 Report

This study develop a force-position compensation control for the feeding drift of bone drilling. The stiffness models of robot and drill-bit are applied. Here are some comments for the authors:

1. Force feedback is quite important factor in this study. This parmeter should be discussed more. Does the time delay of the deformation of the sensing element affect the control algorithm?

2. This study should provide the finite element simulation to verify the model first. 

3. Bone is a kind of biological material. The viscoelestic parameter should be considered. The additional bone model is necessary in this study if the author insisted on the application of bone drilling.

4. minor comments: a. the methods and results should be defined carefully, some results appear in Fig. 4, and b. the euquipment setup appears in the results. c. Besides, the conclusions should be clear and simple.  Too many sentences is bulky and easy to miss the contribution. d. the thickness"m" in line 111.

This research is valuable, but the paper can be improved well. The revision is suggested.

none

Reviewer 3 Report

Review of the paper:

Force-Position Hybrid Compensation Control for Path Deviation in Robot-Assisted Bone Drilling

Authors:

Shibo Li, Xin Zhong, Yuangyuan Yang, Xiaozhi Qi, Ying Hu, Xiaojun Yan

The paper submitted for review requires major corrections to meet the conditions for a substantial scientific publication in the Sensors Journal. The research problem addressed in the article is interesting from an application point of view. However, the organisation of the paper, the development of the results and the general assumptions for the concept of using an industrial robot require clarification and thorough justification. List of detailed comments below:

1)    The Abstract section: authors written: … Specifically, the path deviation is significantly reduced by 56.6%, the force of the tool is reduced by 38.5%, and the hole quality is substantially improved… - What is the quality indicator for making a bone hole? What are the quality parameters?

2)    Lines 14-15 authors written: … significantly improve the accuracy and safety of robot-assisted bone drilling… - Please specify accuracy in relation to a specific medical procedure, as otherwise, this statement is unsupported. In most cases, drilling a hole in the bone is not subject to the constraints of high accuracy that are associated with traditional machining of, for example, metal for industrial purposes. What is the indicator for determining safety when using an industrial robot designed for industrial and not medical purposes? The UR5 robot used is not safe for medical procedures because it is designed to work as a cobot on a production line, not in a hospital room.

1)    The Introduction section: When reviewing the state of the art, authors are encouraged to review papers published in different scientific centres around the world (Europe, Australia, USA, etc.). Please add any other results related to the topics covered and complete this section.

2)    Figure 1 – How can the quality of the hole that is made be determined based on this photo?

3)    The section Materials and Methods: the authors don’t write in this section about the structure and mechanical parameters of human bones. Different skeletal bones will have different structures. Please be specific, this is what we want to drill. Without this, there is no point in modelling the process.

4)    Figure 4 and 5 should be moved to the Results section.

5)    Section 2.2.1. : The adopted manipulator stiffness model does not take into account the elasticity and damping effects of the elements of the kinematic pairs. In this section, please provide an actual mathematical development of the presented model, not just a general formula.

6)    The manuscript does not accurately describe the parameters and type of sensors used in the study, but the paper is scheduled for publication in Sensors Journal. What is the version of the UR5 robot? The new versions of the UR5 have an integrated force and torque sensor.

7)    How were the PID controller settings selected and tuned? The authors do not say anything about this in the article.

8)    Does the position of the NDI optical tracking system relative to the robot affect the uncertainty of the robot end effector position and orientation measurements?

9)    Section 3.2. – Authors don't write about the statistical analysis of the results obtained. After all, every scientific result is estimated with a certain probability.

10)  The authors did not check how the compensation control works for different tool inclinations about the bone surface. Does the compensation also work in dynamic conditions or were the tests only carried out on a static, stationary object?

11)  Figure 11 – indicates that a PID controller was not used in the tests, but only a proportional P controller, since only the Kp setting is changed.

12)  Figure 13 – Please use the quality parameters when evaluating the drill holes. How many samples were taken for quality evaluation? How does drill bit wear affect the deterioration of the hole?

13)  To evaluate the positioning accuracy of the robot with the drilling tool, is the accuracy of the NDI tracking device sufficient?

14)  Often authors use the word weight when they should use mass. Please correct this.

15)  The Conclusion section: authors wrote about drilling path prediction (line 399) and in all the text of the article only Figure 9 tries to explain how this control strategy works. It's not enough. Please elaborate on this point.
